# An Overview of Thermal Treatment Emissions with a Particular Focus on CO$_2$ Parameter

**Deborah Panepinto** * , **Marco Ravina** and **Mariachiara Zanetti** *

DIATI, Department of Environment, Land and Infrastructure Engineering (DIATI), Politecnico di Torino, Corso Duca Degli Abruzzi 24, 10129 Torino, Italy

* Correspondence: deborah.panepinto@polito.it (D.P.); mariachiara.zanetti@polito.it (M.Z.);
  Tel.: +39-011-090-7660 (D.P.); +39-011-090-7696 (M.Z.)

**Abstract:** Waste-to-energy (WtE) technologies can offer sustainable solutions for waste that cannot be further reused or recycled, such as the part of municipal solid waste (MSW) that is not suitable for recycling processes. The two main (most widely used) thermal treatment technologies that can be applied to MSW are direct combustion in an incineration plant and gasification. This paper examines in particular the direct combustion in incineration plants, explaining the main process, the main technologies applied, and the resulting environmental aspects. Moreover, this work focuses on analyzing flue gas emissions from thermal treatment in order to better understand the impacts of these kinds of processes. A particular focus on the CO$_2$ parameter is performed. CO$_2$ is a persistent atmospheric gas, and it is one of the greenhouse gases (GHGs) potentially responsible for the climate change phenomenon. In this sense, specific indexes (tCO$_2$/tMSW and tCO$_2$/MWh) are elaborated considering the thermal treatment plants present in six Italian regions. The main aim of this review paper is to try to fill the gap that still exists regarding the emissions environmental compatibility coming from these type of plants, the evaluation of the amount of CO$_2$ emitted, and the possible reduction of the CO$_2$ parameter. One of the main outcome obtained is in fact the evaluation of the amount of CO$_2$ coming from these kinds of plants and some indications about the technological possibilities of reducing this amount.

**Keywords:** MSW thermal treatment; incineration; gasification; flue gas emissions; CO$_2$ emissions

## 1. Introduction

Municipal solid waste (MSW) is a problem that affects the entire world. The Waste Framework Directive (WFD)—Dir. 2008/98/EC governs waste management in the European Union [1]. This Directive is codified in the Consolidated Environmental Act (TUA, Italian Legislative Decree 152/2006) [2] under Italian law. The WFD is based on various waste management ideas, including:

- Reducing resource consumption;
- Taking into account the complete life cycle of materials/products;
- Achieving the best environmental outcomes overall;
- Using the expanded producer responsibility system to implement the "polluter pays" idea.

The hierarchy of operations that the Directive defines for waste mechanism is as follows:

- Prevention;
- Reuse;
- Recycling;
- Recovery (i.e., energy recovery);
- Final disposal in a landfill.

This hierarchy specifies the priority for the application of management operations, according to which prevention of waste generation is preferable to any other management

method. Furthermore, since prevention is part of the production process of the waste, the first real management intervention is the second priority, namely, reuse.

In fact, when residue has been produced, the first choice is to reuse the material (in the same process that produced it), and the second is to recycle it after a treatment. If neither of these options is sustainably viable, then it is preferable to use the waste material for recovery (such as energy recovery in thermal treatment plants) before sending it for final disposal in a landfill.

The WFD also specifies that it is possible to deviate from this general hierarchy in particular cases if a greater environmental benefit can be demonstrated on that basis, for example, via a Life Cycle Assessment (LCA).

The last amendment to the WFD was introduced at the end of May 2018 with the introduction of the so-called "Circular Economy Package".

The "Circular Economy Package" reinforces some concepts that were already contained in the WFD and also clarifies some definitions in order to improve the collection and processing of statistical data on waste management. Particular attention is paid to the target of keeping materials within the cycle of production and consumption as long as possible so as to minimize the need for virgin materials and the amount of waste to send for final disposal.

From the viewpoint of average MSW production in European countries, the amount is more or less stable throughout the year, as can be seen in Figure 1, which depicts the production of MSW in EU countries for the two years of 2005 and 2020 (the data came from [3]).

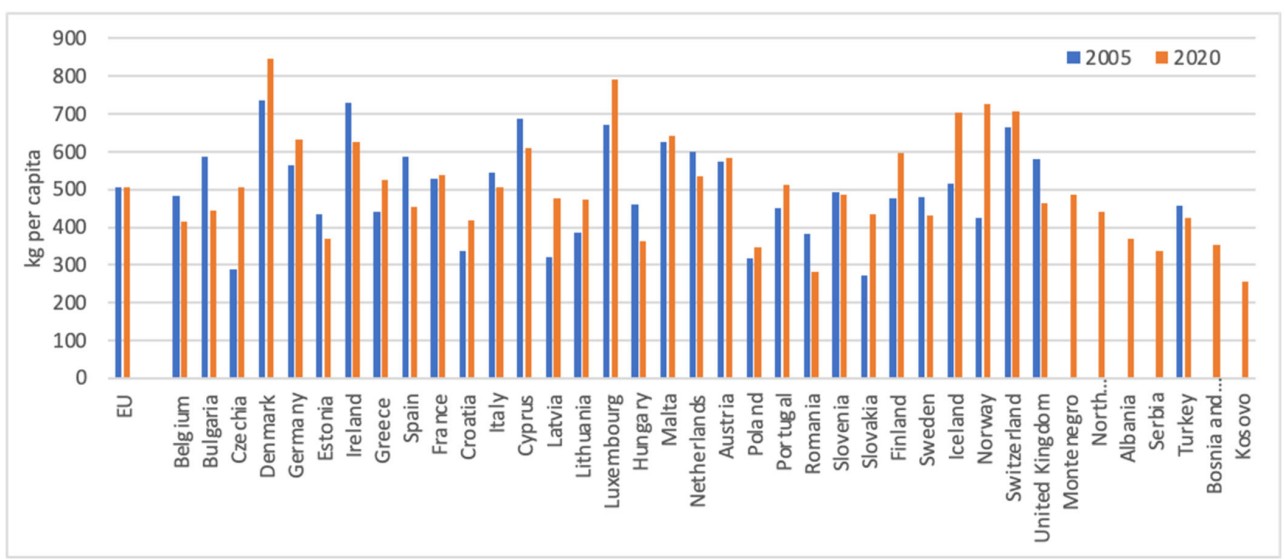

**Figure 1.** MSW generated in the EU in the years 2005 and 2020.

Figure 2 shows the MSW management situation in the year 2019 in EU countries (graph elaborated used the data reported in [4]). Here, it is possible to see deep differences between the most advanced countries (on the left), where final disposal in landfills has been almost completely eradicated, and the other countries (on the right) where the landfill method is still widely used.

The shift to environmental, economic, and social sustainability depends on effective management. It is important to note that environmental and social sustainability (ESS) is the adaption and integration of precautionary environmental and social principles and considerations into decision-making processes. The paradigm for waste management is evolving as a result of this shift; although waste is solely seen as a burden in a linear economy, it may be seen as a resource in a circular economy (and this is, for example, a solution to also improve the ESS). The annual amount of MSW generated worldwide is

2.01 billion tons, and by 2050, that amount is predicted to increase to 3.40 billion tons [1]. As a consequence, the production of energy from waste that cannot be reused or recycled can represent a solution in line with the principles of the circular economy and can contribute to energy diversification [2,5]. The waste-to-energy (WtE) process can currently be achieved by several different technologies, such as anaerobic digestion, the production of waste-derived fuels, (co-)incineration in combustion plants and in cement and lime production or in dedicated facilities, or indirect incineration following a pyrolysis or gasification step. Among the WtE technologies, incineration is the most-established process, accounting for more than 1400 plants worldwide [3]. Incineration or "direct combustion" is the complete, rapid exothermic oxidation of the waste organic fraction in the presence of an adequate excess of oxygen. Incinerators work with many different types of waste, including MSW [4–7], products discarded after the completion of their use phase (such as end of life tires, [8]), solid refuse fuels (SRFs) [9,10], industrial waste (IW) [11,12], and industrial hazardous waste (IHW) [13]. In addition to being a solution for waste management, incineration provides heat and can generate steam and electricity [14]. Gasification, often known as "indirect combustion", is another WtE process that involves the thermochemical decomposition of MSW to produce combustible gas (syngas) and a later combustion step for energy recovery (two-step oxidation) [5,15]. Germany and Italy have the most gasification facilities, but the Scandinavian nations have the largest individual plants, according to a European project report [16]. A thermochemical conversion process known as gasification can handle a wide range of feedstocks, including biomass, municipal solid waste, and other solid waste [17]. The organic content of the waste is converted mainly to carbon monoxide and hydrogen, along with lower amounts of methane, although syngas is generally contaminated with undesired products such as particulates, tar, alkali metals, chloride, and sulphide. The obtained syngas, though, can be used to produce chemicals (such as fertilizers [18]) and fuels [19,20], or for power generation [20,21].

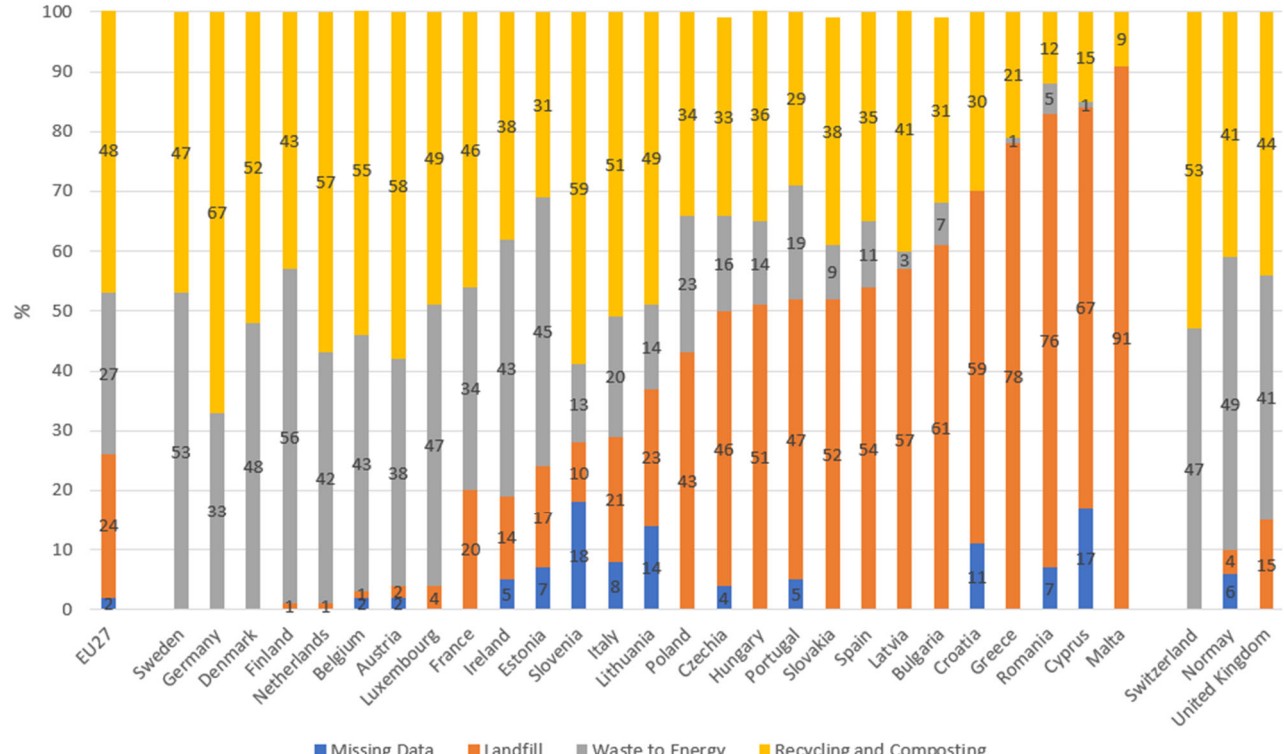

**Figure 2.** Municipal waste treatment in 2019 (EU 27 + Switzerland, Norway, and United Kingdom).

Air pollutants such precursors of $SO_X$, $NO_X$, HCl, particulate matter, components of hydrocarbon, dioxins, and $CO_2$ are direct emissions from thermochemical conversion tech-

nologies [4,22]. However, specialized flue gas treatment methods are able to significantly reduce air pollution from these emissions [23].

Thus, as indicated, gaseous emissions represent some of the main, potential, environmental (and human health) impacts of energy-intensive industries. The flue gas is treated in order to significantly lower the amounts of principal contaminants before being released into the atmosphere (macro and micro pollutants). Due to the increasingly stringent regulations and concrete technological advancements, which have resulted in the development of highly sophisticated systems that enable emission values to be achieved at the upper limit of the measurable threshold, the flue gas depuration line is very articulated and complex [7,24,25].

In addition to the release of primary pollutants, thermal treatments plants are responsible for the release of greenhouse gases (GHGs), in particular $CO_2$, into the atmosphere, which are themselves responsible for the phenomenon of climate change. Unlike what happens with primary pollutants for $CO_2$ there are no consolidated abatement/reduction technologies on an industrial scale and no limit concentrations to be respected [26,27].

In this review, the flue gas emissions from thermal treatment are analyzed in order to better understand the potential environmental impact of these kinds of processes. In the first part, a complete review of the thermal treatment applied to the MSW matrix was performed. In the second part, the residual pollutant concentration (after the flue gas depuration system) was analyzed and compared to other industrial plants and with the current legislation.

Furthermore, a particular focus on $CO_2$ parameter was performed. $CO_2$ is a persistent atmospheric gas, and it is one of the greenhouse gases (GHGs) potentially responsible for the climate change phenomenon. As previously indicated, for $CO_2$ there are no consolidated reduction technologies and no limit concentrations to be respect. Moreover, generally the concentration of these parameters are not measured (because there are not limited concentrations to respect). In the literature there are some papers that estimate these amounts [28–33]. In order to limit these gaps in this review, a specific index ($tCO_2/tMSW$ and $tCO_2/MWh$) was elaborated considering the thermal treatment plants present in six Italian regions, and then some considerations, using the literature, concerning the possible reduction of these parameters were reported.

## 2. Thermal Treatment Approach

Thermal treatments are high-temperature chemical operations that break down organic materials to create new chemicals with a more straightforward chemical composition [34,35]. Any thermal treatment's main objective is to change wastes into compounds that are less harmful to the environment and human health in order to reduce the volume and quantity of waste that must be dumped in landfills while also recovering energy [36] (Figure 3).

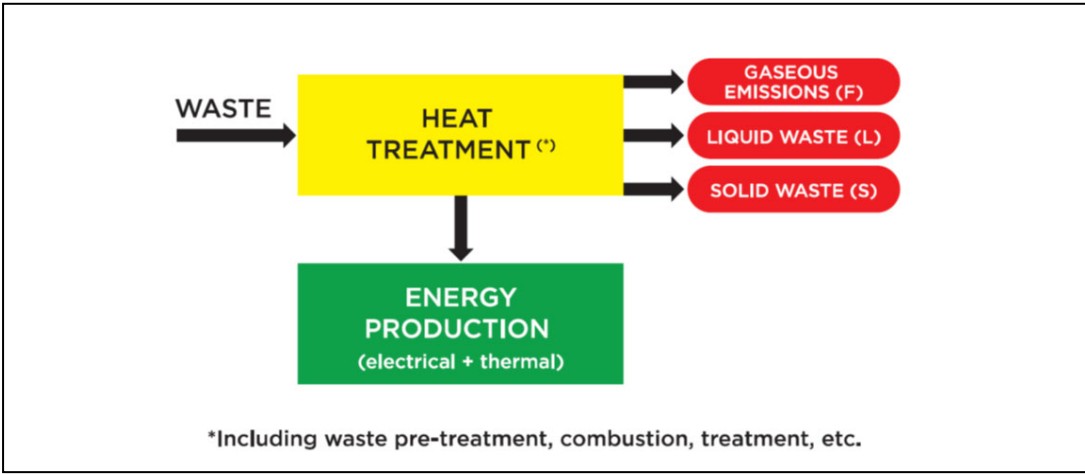

**Figure 3.** Simplified diagram related to heat treatment.

In the waste sector, the following thermal treatments can be applied:

- Direct combustion in an incineration plant;
- Gasification;
- Pyrolysis.

Among these, the first one—incineration—has so far been the one most frequently applied to solid waste; experience with incineration on an industrial scale is now very extensive.

The three primary thermal treatment procedures can be schematically illustrated as shown in Figure 4 if we define R as the ratio between the actual (stoichiometric) amount of the oxidizing agent (air and/or oxygen employed) and the theoretical amount.

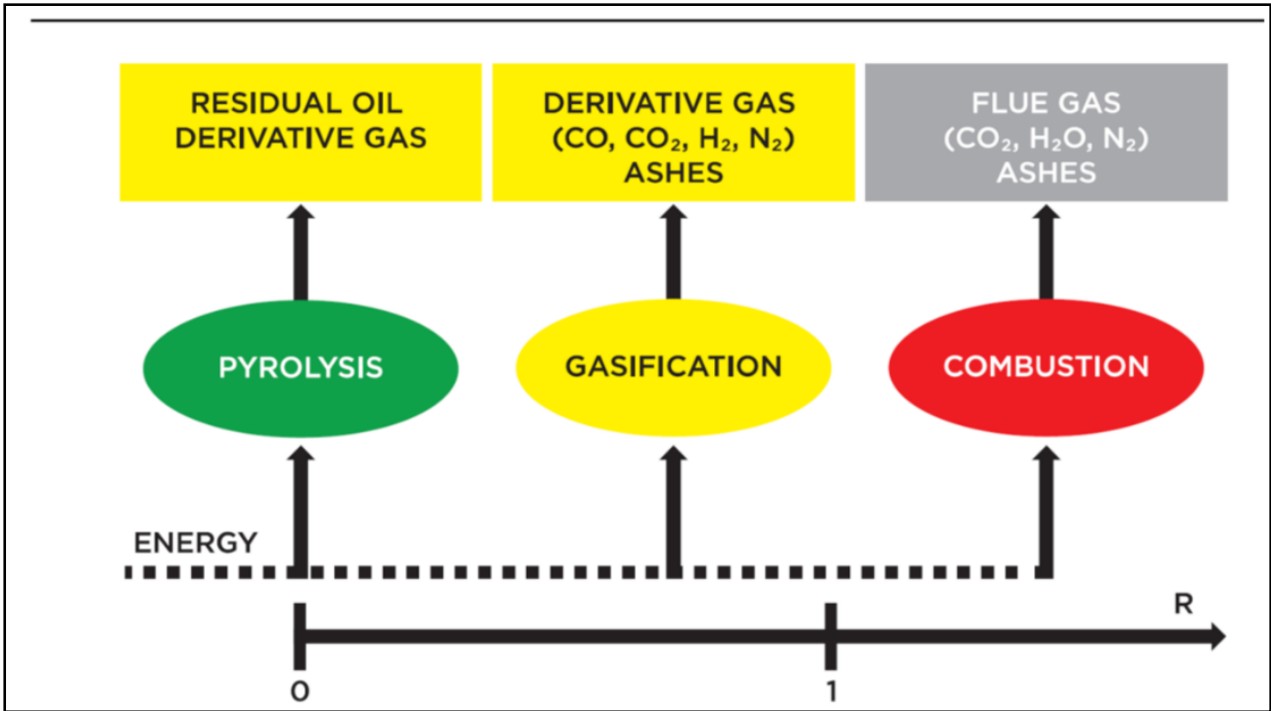

**Figure 4.** Schematic representation of thermal treatments.

Thus, direct combustion in an incineration plant presents an R higher than 1, so the direct combustion happens with an amount of oxidizing agent higher than the stoichiometric amount; in fact, direct combustion (incineration) is the complete oxidation of the MSW in the input (the main aim of the excess of air is to guarantee the complete oxidation of the organic substance). Gasification presents an R lower than 1, so gasification is a partial combustion of the MSW in the input (the amount of oxidizing agent used during the process is lower than the stoichiometric amount). Finally, pyrolysis presents an R equal to 0, so pyrolysis can be defined as thermal degradation of the MSW in the input (without an oxidizing agent).

The thermal oxidation technique (combustion in particular) used to burn MSW produces simple molecules in a gaseous form by oxidizing the fundamental elements that make up the organic compounds (flue gas). The inorganic portion of the waste is also oxidized and exits the process as a solid residue that can be disposed of or reclaimed. The organic carbon is converted to carbon dioxide ($CO_2$), hydrogen to water ($H_2O$), sulfur to sulphur oxide ($SO_2$), and so on. Since the process is oxidative, oxygen must be present for the reactions to occur. Air is often used, and it is supplied in excess of the stoichiometric amount to facilitate chemical processes, as previously reported [36].

Figure 5 shows the amount of MSW incinerated in European countries in the year 2016 (the data came from [37]).

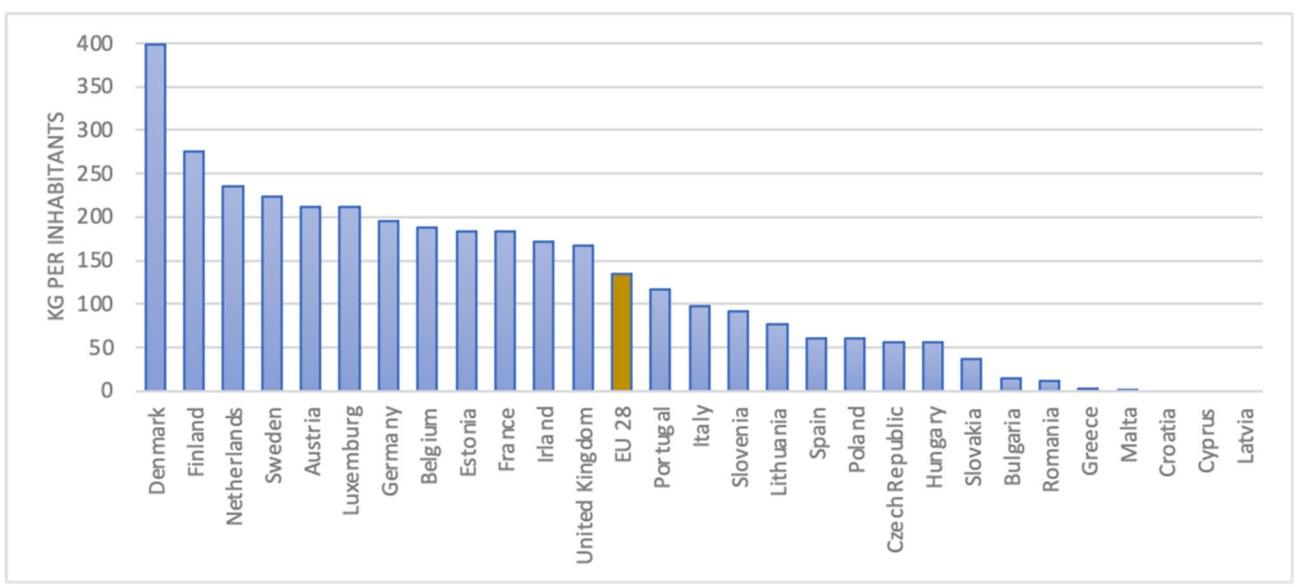

**Figure 5.** Per-capita incinerated MSW in Europe in 2016.

The main types of thermal treatment (incineration, gasification, or pyrolysis) plants are [36]:

- Combustion chamber. The most common combustion technologies for MSW treatment are the moving grate system and the fluidized bed system. The first one (moving grate system) is the most widely used technology on account of its flexibility of operation and reliability resulting from extensive application. The technology consists of a grid, inclined to a horizontal line, which is covered with a waste bed that is several dozen centimeters thick. The grid is made up of a number of "fire bar" components that are arranged to allow the combustion air to pass through and be distributed across the entire waste bed. Both directly inside the combustion chamber and underneath the grid, the combustion air is injected. The residence time of the waste in the combustion chamber (and thus on the grid) is generally between 30 and 60 min (in order to allow for the complete oxidation of the organic substances). With the proper systems, the remaining bottom ash produced by the operation is released from the grid's final section (water bath accumulation tanks). In accordance with an adequate oxygen content (6–8%) and turbulence, temperatures in the range of 950–1000 °C are thought to be sufficient to ensure the complete oxidation of the organic components.

The fluid-bed furnace consists of a combustion chamber with a specified amount of inert material (the "bed") stored inside. The "bed" of the fluid-bed furnace is often made out of sand that is suspended in an upward air current (which also acts as a comburent). Consistent and full combustion is made possible by the movement of the grid bed, which also contributes to significant temperature uniformity and mixing. The equipment is designed for the burning of relatively homogeneous and tiny chemicals or particles and was initially used in the petrochemical industry (such as, for example, wastewater sludge). Therefore, MSW must go through at least one shredding process to reduce the size of its particles.

At the end of the thermal treatment process, two different types of solid residue are generated:

○ Bottom ash, generally equal to 20% of the waste input to the plant. This type of residue is generally sent to recovery;
○ Fly ash, generally equal to 4–7% of the waste input to the plant. This type of residue, removed through the flue gas depuration line, is a hazardous waste and is generally sent to a landfill for hazardous waste.
- Flue gas depuration line. A thermal treatment plant generates three different outputs: gaseous, liquid, and solid emissions. Before being released into the atmosphere from

the chimney, the flue gas must be treated in order to reduce the concentrations of the pollutants generated during the process. The flue gas treatment section is extremely articulated and complex as a result of the increasingly strict regulatory limits placed upon it and of actual technological progress, which in recent years has resulted in the development of sophisticated systems capable of producing emissions much lower than the maximum permitted by legislation.

○　The pollutants present in the flue gas can be grouped into two different types:

○　Macro-pollutants: substances present in the flue gas in concentrations in the order of $mg/Nm^3$, such as PM (particulate matter), sulphur oxides (generally $SO_2$), nitrogen oxide ($NO_x$), carbon monoxide (CO), and halogen acids (generally HCl and HF);

○　Micro-pollutants: substances present in the flue gas in much lower concentrations (in comparison with the concentrations of macro-pollutants), which include both inorganic species, such as heavy metals (Cd, Cr, Hg, Pb, Ni, and so on) and organic species, such as PCDD, PCDF, and PAHs.

With regard to the reduction of these pollutants (both macro- and micro-), according to current legislation it is necessary to use the best available techniques (BATs) defined and reported in the official document of the IPPC Bureau for incineration plants [37].

-　Energy recovery section. Energy recovery from incineration is commonly achieved by capturing the steam emitted when the flue gases are cooled—a process required for their subsequent treatment.

Energy recovery from gasification and pyrolysis, by contrast, is generally obtained from the steam emitted from the combustion of the syngas generated during these processes. There are two possible configurations of such a plant: an electricity-only configuration (i.e., the production of only electrical energy) or a cogenerative configuration (the production of both electrical and thermal energy). Generally, in the electricity-only configuration, the gross yield is about 30%, while in the cogenerative configuration the gross yield (from electricity and thermal revenue) is about 70%.

## 3. Environmental Aspects

Flue gas is the output from the thermal treatment chimney, and in particular the residual pollutant concentrations contained represent one of the main negative environmental impacts.

In order to ensure that the residual pollutant concentration in the flue gas output from the chimney is lower than the maximum value allowed by the national regulations, it is necessary, as previously indicated, to use the best available techniques (BATs) included in the BAT conclusion document, "Commission Implementing Decision (EU) 2019/2010 of 12 November 2019 establishing the best available techniques (BAT) conclusions, under Directive 2010/75/EU of the European Parliament and of the Council, for waste incineration" [38].

Figure 6 provides an overview of the performances anticipated from the existing treatment systems introduced by the most recent iteration of the BAT conclusion document. The emission limits found in the European rules for the industry are also provided for comparison. With reassuring margins of compliance for some of the contaminants of greatest concern—"first and foremost" dioxins and hazardous metals—the current plant framework demonstrates a situation that is completely compatible with the mandated limits. Remember that all of this is happening in the face of regulations that are now among the strictest in comparison to all other emissive sectors, including stationary combustion and industrial operations.

An overview of the most recent measurements for some of the most prominent plants operating in Italy is provided in Figure 7, which supports the comparison of the Italian and European environments. The annual values received through periodic monitoring or continuously recorded by systems and sent to the controlling authorities are displayed in terms of annual values.

| Contaminant | 2010/75/EU, IED | BAT |
|---|---|---|
| Powders | 10 | < 2–5 |
| HCl | 10 | < 2–8 |
| HF | 1 | <1 |
| $SO_2$ | 50 | 5–40 |
| NOx (as $NO_2$) | 200 | 50–150 (180 without SCR) |
| TOC | 10 | < 3–10 |
| CO | 50 | 10–50 |
| Hg | 0.05 | 0.001–0.02 |
| Cd + Tl | 0.05 | 0.005–0.02 |
| Other metals | 0.5 | 0.01–0.3 |
| PCDD/F ($\eta$gTEQ/m$^3$) | 0.1 | < 0.01–0.08 |
| $NH_3$ | – | 2–10 |
| PAHs ($\mu$g/m$^3$) | 10 | – |

**Figure 6.** Industrial Emissions Directive's 2010/75/EU emission ceilings and the emission ranges associated with BAT.

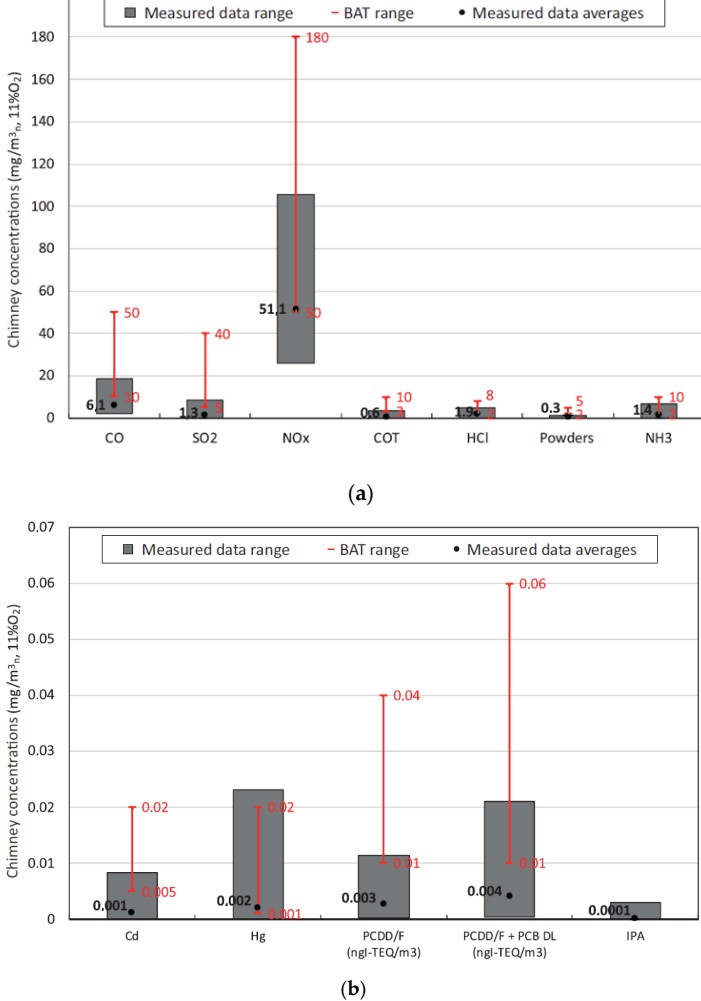

**Figure 7.** Emission measurements from Italian facilities for the three years between 2016 and 2018 are compared to the industrial BAT values for conventional (**a**) and trace hazardous (**b**) pollutants.

Another element of significance to the inventory data is the comparison with emissions from other activities, which are frequently significant contributors to air quality. In order to do this, Table 1 highlights the image of the function of the sectors that frequently overlap with incineration, either due to their contextual presence in the area or due to similar energy production aims, as indicated by the data from the most recent national inventory, referring to 2018.

**Table 1.** Annual emissions from Italy's major industries of the contaminants of greatest concern between 2000 and 2018 (processed from ISPRA data, 2020).

| 2000 | Residential and Commercial Combustion | Energy Production and Distribution | Combustion in Industry | Production Processes | Road Transportation | Waste Incineration |
|---|---|---|---|---|---|---|
| $SO_2$ | 3.5% | 66.4% | 14.2% | 3.4% | 1.6% | 1.3% |
| $NO_x$ | 11.7% | 11.6% | 12.2% | 0.4% | 50.6% | 0.16% |
| $PM_{10}$ | 35.0% | 8.1% | 8.6% | 7.2% | 21.2% | 0.01% |
| CO | 22.1% | 1.2% | 6.7% | 2.6% | 63.5% | 0.002% |
| Cd | 25.0% | 0.0% | 62.5% | 12.5% | 0.0% | 2% |
| Hg | 7.7% | 46.2% | 23.1% | 23.1% | 0.0% | 1% |
| Pb | 2.7% | 0.5% | 16.0% | 6.9% | 72.5% | 0.3% |
| PCDD/F | 41.7% | 2.2% | 22.0% | 29.9% | 4.2% | 5.3% |
| PAHs | 79.6% | 4.0% | 0.0% | 12.6% | 3.3% | 0.1% |
| 2018 | Residential and Commercial Combustion | Energy Production and Distribution | Combustion in Industry | Production Processes | Road Transportation | Waste Incineration |
| $SO_2$ | 9.4% | 33.3% | 24.0% | 12.4% | 0.4% | 1.2% |
| $NO_x$ | 13.0% | 7.0% | 9.4% | 0.8% | 43.5% | 0.8% |
| $PM_{10}$ | 53.8% | 1.0% | 4.7% | 9.3% | 11.8% | 0.02% |
| CO | 61.9% | 1.9% | 4.1% | 3.6% | 19.9% | 0.04% |
| Cd | 9.4% | 3.3% | 38.1% | 29.1% | 7.7% | 1.2% |
| Hg | 7.0% | 19.3% | 27.4% | 43.0% | 2.6% | 2.6% |
| Pb | 6.8% | 1.1% | 44.8% | 40.6% | 5.1% | 2.7% |
| PCDD/F | 37.5% | 1.7% | 20.2% | 32.1% | 3.8% | 0.2% |
| PAHs | 78.1% | 0.7% | 0.8% | 13.9% | 3.8% | 0.007% |

The estimations confirm a very small, if not nearly negligible, contribution of incineration emissions compared to those from all other sources, without any pretense of generalizing the national structure to territorial and productive contexts localized in constrained areas. The archive reveals a significant prevalence of residential and commercial combustion for conventional contaminants, particularly for dust (nearly 60%) and CO (approximately 64%), which also affect several trace species, particularly PAHs, though with less reliable estimation. As in the past, it was verified that road transportation, primarily from diesel engines, is the main source of $NO_x$ [36].

Dioxins in particular, whose main origins might be associated with the industrial sector (combustion and production processes), as well as civil stationary combustion, present a similar situation when it comes to incineration for both trace metals and organic micro-contaminants.

In order to better understand the role of thermal treatment plants on air quality, the example of the Turin (TRM, Gerbido, Turin, Italy) incineration plant is presented. The facility, which commenced operations in 2014, has been given permission to handle the municipal solid waste (MSW) portion that is left over after separate collection [38,39]. The lower heating value (LHV) of this residual fraction is in the range of 11 MJ/kg.

The plant contains three equal lines, each of which has a combustion zone, an energy recovery section, and lastly a section for flue gas treatment. The plant runs for 7800 h each year. A boiler and a steam turbine are included in the energy recovery portion, while the combustion zone uses a moving grate mechanism. The generation plant runs in a cogenerative mode at the moment. A dry scrubber (with the addition of sodium bicarbonate and activated carbon) for the removal of acid gases as well as organic and inorganic micro pollutants, a bag filter for the removal of residual and generated dust,

and a final selective catalytic system for the reduction of NOx make up the section for the treatment of the final flue gas [39]. By combining the aforementioned treatments, it is possible to ensure that the concentration of pollutants in the output flue gas meets the threshold values established by the national law for the release of gaseous effluents into the atmosphere (D. Lgs. 152/2006, BATC 2019) (Figure 8).

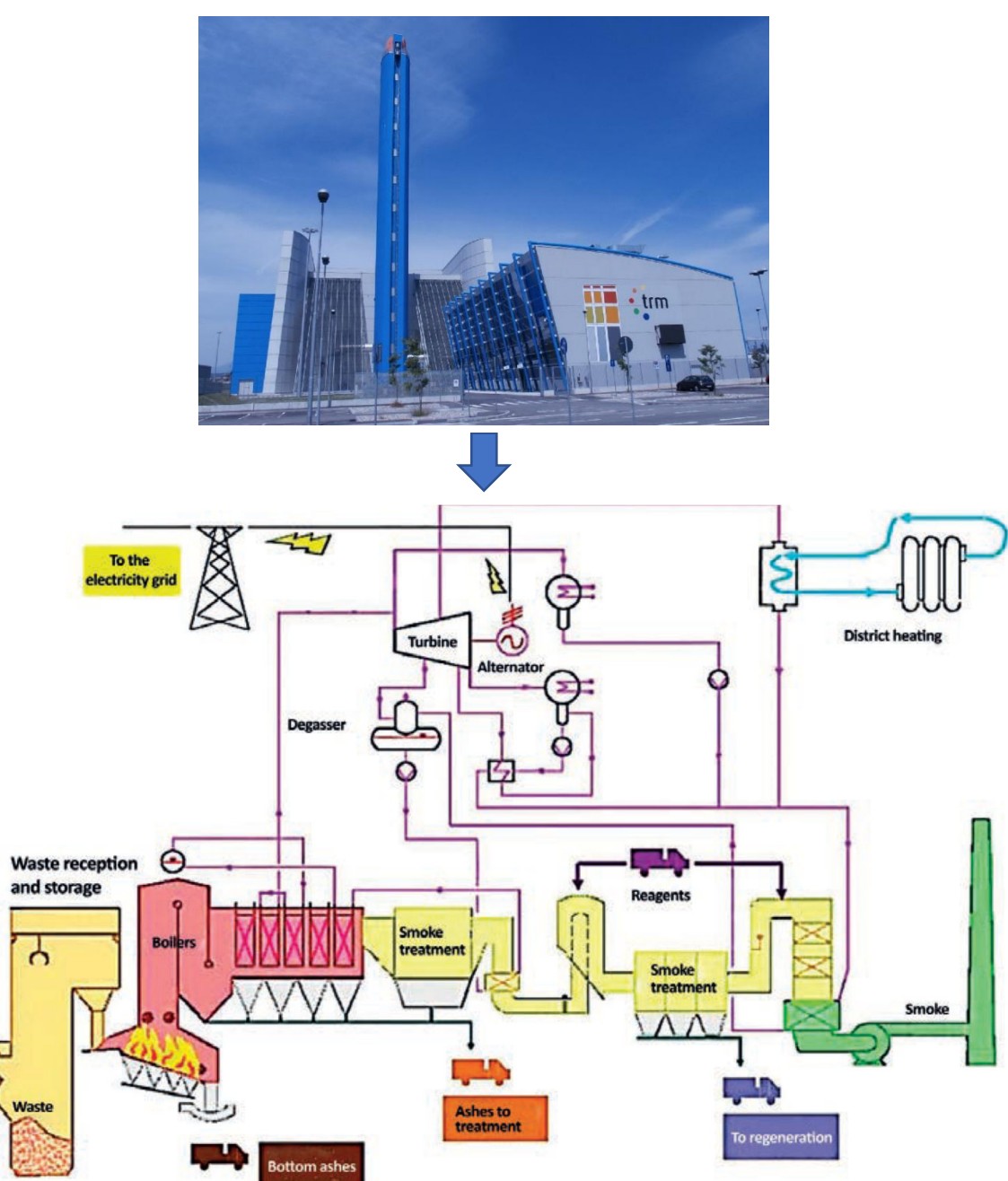

**Figure 8.** Turin incineration plant scheme.

Figures 9 and 10 report the real-time measures (for certain days in May and June 2022) for the two pollutant parameters, NOx and dust, emitted from the chimney of the Turin incineration plant. As can be seen, the concentrations of these two parameters were always lower than the authorized limit.

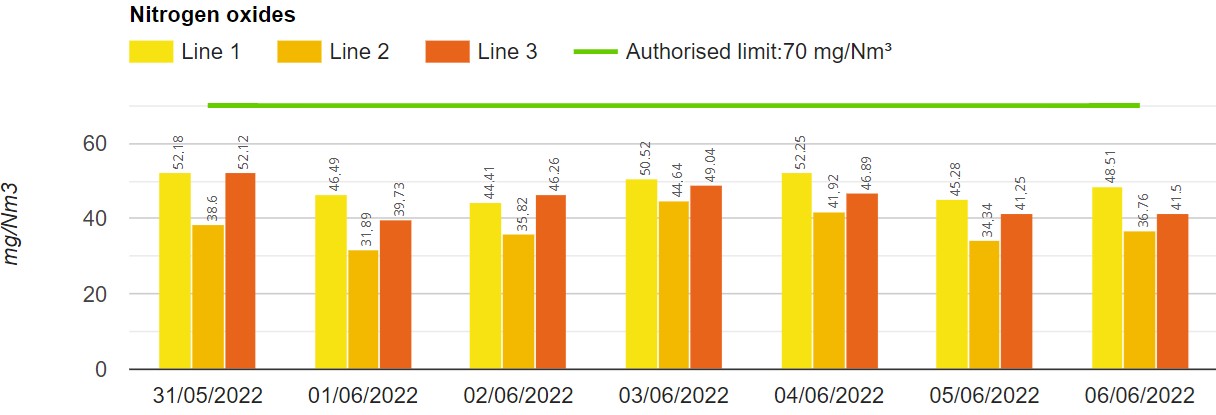

**Figure 9.** Turin incineration plant real-time emissions: $NO_x$ parameter.

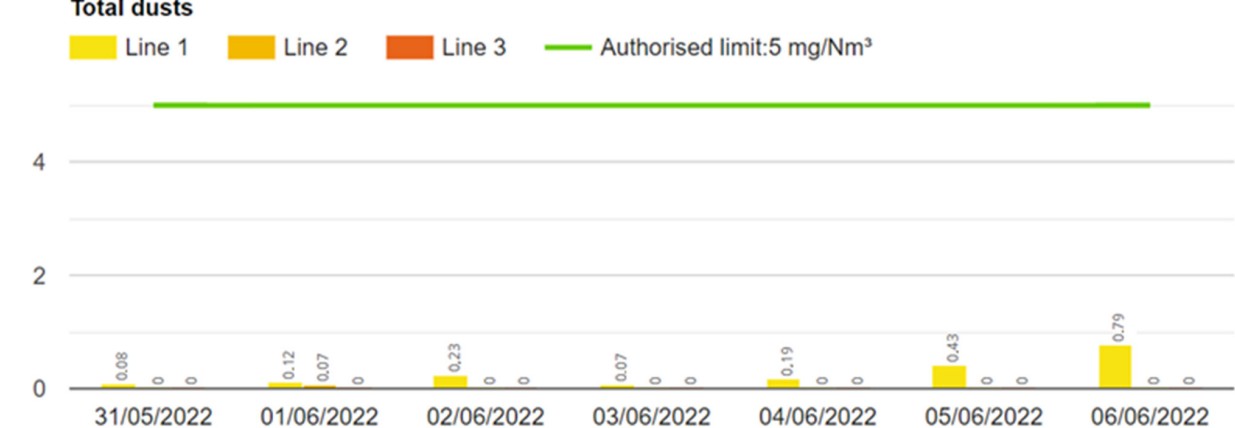

**Figure 10.** Turin incineration plant real-time emissions: Total dust parameter.

### 3.1. $CO_2$ Emissions

Environmental contamination is unquestionably a bad externality, as we all know. It is crucial for authorities to create appropriate environmental policies as a driver through which any government can exert environmental governance in order to deal with this externality and reduce the carbon emissions from the activities of governments, businesses, and individual citizens [40]. Environmental policy rigor and regulations are another strategy that can be used to combat environmental degradation. The goal of stringent environmental policy is to increase the cost of climate services and pollution in order to change consumer and producer behavior and encourage the creation and use of environmentally friendly products [41]. This might be accomplished by placing restrictions on the contaminating agents, which will increase the expense of activities that produce pollution and lessen their allure [41,42].

Thus, as previously indicated, in addition to the release of primary pollutants, industrial plants are responsible for the release of greenhouse gases (GHGs) into the atmosphere, which are themselves responsible for the phenomenon of climate change. Unlike what happens for primary pollutants for GHGs, there are no consolidated abatement/reduction technologies on an industrial scale and no limit concentrations to be respected.

The interactions in the atmosphere between different pollutants, between pollutants and greenhouse gases, and between pollutants and meteorological variables affected by climate change, as well as interactions with terrestrial ecosystems and the chemistry of the pollutant transformations in the atmosphere as a function of the climatic regime, are all topics of the most recent scientific debate. This relationship between climate change and air quality is very complex. The linkages and synergies between these many processes, as well as the physico-chemical mechanisms describing how air quality and climate change

interact and influence one another, are not fully understood. Even the modeling used to create future situations and their effects on air quality is occasionally unable to adequately depict them.

From the point of view of the influence of climate change on climate, it should be noted that the increase in greenhouse gas concentrations alters the radiation balance between the atmosphere and the surface of the earth, leading to a change in environmental conditions, including the increase in temperatures, the increase in meteorological regimes capable of determining changes in the chemical transformations that occur in the atmosphere, and, therefore, in the chemical composition of the atmosphere itself. In particular, an increase in temperatures and irradiation conditions could increase the concentrations of ground-level ozone and secondary pollutants [27].

It is important to emphasize that when considering strategies to slow down climate change and simultaneously improve air quality, it is important to assess the effects of both these phenomena. As a result, one should look for synergies (win–win) and steer clear of solutions that would worsen one of the two phenomena (win–lose). To be successful, it is therefore necessary that the reduction policies are integrated and evaluated in a combined way, including reciprocal compensations [43,44].

### 3.1.1. $CO_2$ Emissions from Direct Combustion Incineration Plants

As carbon dioxide is a product with a major greenhouse effect, the calculation of its production is essential for evaluating the impact on climate change. As previously indicated, this parameter plays an important role in thermal treatment processes because to date no consolidated abatement technologies have been developed. It is therefore assumed that all the $CO_2$ generated during the thermal treatment process is emitted through the plant chimney.

In order to have an idea of the $CO_2$ quantities emitted by these type of plants, six Italian regions were evaluated where the incineration processes are implemented. These regions are Piedmont, Lombardy, Trentino South Tyrol, Veneto, Friuli Venezia Giulia, and Emilia Romagna. The six chosen regions are ubicated (as shown in Figure 11) in the north of Italy. They were chosen in order to have rather homogeneous characteristics in the composition of the waste in order to be able to perform comparisons among the obtained results.

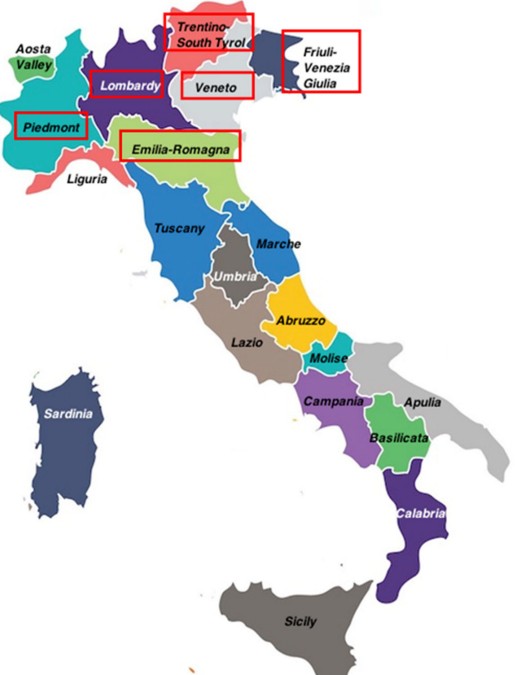

**Figure 11.** Ubication of the analyzed regions (highlight in red).

For these regions, the amount of MSW incinerated and the amount of energy recovery was considered. Starting from these data, the amount of $CO_2$ (in terms of the two indexes $tCO_2/tMSW$ and $tCO_2/MWh$) was evaluated.

In order to perform a carbon dioxide balance, we proposed the following considerations:

- It was assumed that all the carbon contained in the input waste stream would become carbon dioxide; hence, the $CO_2$ was calculated by the following equation:

$$CO_{2produced} = C_{total} \times (MOL\ WT\ CO_2/MOL\ WT\ C) \times Waste\ Flow \tag{1}$$

where $C_{total}$ = percentage of carbon in the original waste, MOL WT $CO_2$ = carbon dioxide molecular weight, MOLWT C = carbon molecular weight.

In order to evaluate the percentage of carbon in the original waste, the tool of mass balance was used, starting from the composition of the MSW (and the elemental composition of the single materials) in the input to the considered thermal treatment plants.

In order to obtain all the necessary data, some spot investigations and analysis were performed.

- It was assumed that the $CO_2$ avoided was considered in the total energy recovery from the thermal treatment; hence, it was calculated by the following equation:

$$CO_{2avoided} = [(Power_{el} \times EF_{el}) + (Power_{th} \times EF_{th})] \times availability\ (h/y) \tag{2}$$

where $EF_{el}$ = emission factor for the electric energy generation, $EF_{th}$ = emission factor for the thermal energy generation; an emission factor (EF) is defined as the weight of pollutant issued by a source referred to the entity of the energetic production (MJ, kWh).

In Table 2, the main obtained results are reported.

**Table 2.** Results of carbon dioxide balance.

|  | Piedmont | Lombardy | Trentino Alto Adige | Veneto | Friuli Venezia Giulia | Emilia Romagna |
|---|---|---|---|---|---|---|
| Treated wastes (t/y) | 560,000 | 2,400,000 | 130,000 | 240,000 | 147,000 | 1,100,000 |
| Produced energy (MWh) | 1,443,750 | 4,568,675 | 255,244 | 519,452 | 241,500 | 1,991,139 |
| Produced $CO_2$ (t/y) | 924,000 | 3,792,000 | 209,300 | 379,200 | 236,670 | 1,573,000 |
| $tCO_2$/t wastes | 1.65 | 1.58 | 1.61 | 1.58 | 1.61 | 1.43 |
| $tCO_2$/t MWh | 0.64 | 0.83 | 0.82 | 0.73 | 0.98 | 0.79 |
| Avoided $CO_2$ (t/y) | 518,000 | 1,600,000 | 90,000 | 185,000 | 86,000 | 707,000 |
| $tCO_2$/t wastes | 0.73 | 0.91 | 0.92 | 0.81 | 1.02 | 0.79 |
| $tCO_2$/t MWh | 0.28 | 0.48 | 0.47 | 0.37 | 0.62 | 0.43 |

By analyzing the results, it is possible to note that the elaboration was performed considering both total $CO_2$ produced (without the amount of $CO_2$ avoided following the energy recovery) and the $CO_2$ avoided following the energy recovery performed from the considered thermal treatment processes.

The values of the obtained indexes (in terms of t $CO_2$/t MSW and t $CO_2$/tMWh) were very similar, and the reason is that, as previously indicated, the compositions of the MSW in the six regions analyzed were rather equal. The main differences were due to the different amounts of energy recovery (and so these differences are concerned with in particular the index $tCO_2$/MWh).

As it is possible to see from Table 2 that the emissions of $CO_2$ were equal (considered an average of the six analyzed regions) to 0.86 t/t MSW and 0.44 t/MWh. These values are similar to the results present in the scientific literature [8] and are lower than values obtained with other options of MSW management (for example, considering the landfill option, this index is equal to 3.28 t $CO_2$/t MSW [8]; in this case, in fact, since there is no energy recovery, the index is higher).

### 3.1.2. $CO_2$ Reduction Possibility

For the $CO_2$ reduction, there are some possibilities tested, although only on the lab scale. One of these is the use of $CO_2$ for the production of sodium carbonate.

Sodium carbonate is one of the most used reagents for the purification of the emitted flue gas from incineration plants.

By putting the emitted flue gas in contact with a solution of NaOH, the carbon dioxide present reacts with the base to form sodium carbonate, giving rise to a solution of odorless and colorless $Na_2CO_3$.

$$CO_2 + 2NaOH \rightarrow Na_2CO_3 + 2H_2O \qquad (3)$$

The gas can be flowed through a counter-current absorber column compared to the soda solution, which is expected to be added to the head of the column after having appropriately optimized the operating parameters and dimensioned the system.

The gas then exits from the head of the column after passing through a demister, an item of equipment consisting of a dense metal grid for the removal of droplets of liquid entrained by the gas, while the solution is collected and sent where necessary.

However, the process requires a rather massive use of NaOH, the production process of which also generates $CO_2$; thus, the installation of a system of this kind must be carefully evaluated.

Recently, with the growing urgency to adopt long-term sustainable technological solutions, research has been directed towards processes involving the use of materials of natural origin (not chemically modified) and of low energy consumption. Among the many proposed solutions, low-temperature physical adsorption using materials of natural origin is of particular interest [45].

The most potentially suitable materials for this type of process are natural zeolites, in particular clinoptilotite, chabasite, faujasite, and cowlesite. These materials show an adsorption capacity of around 70–100 g $CO_2$/kg at low pressures and temperatures [46]. Compared to other synthetic or additive materials (e.g., modified zeolites or metallic organic frameworks), natural zeolites have a lower adsorption capacity [47]. On the other hand, the environmental impact and cost of these materials are significantly lower. Natural zeolites show higher adsorption capacities as the partial pressure of carbon dioxide increases and the temperature decreases. A low Si/Al ratio within the material also favors the presence of exchangeable ions within the structure. On the other hand, the use of these materials should be considered with caution if water or other competing species are present within the gas to be treated. In addition, to occupying the adsorption sites, given the strong dipole moment possessed by the molecule, water could distort the electric field generated by the cations within the zeolite, thus decreasing the material's performance [48]. Regarding regeneration capacity, there are still conflicting results in the literature [49].

An energy-efficient adsorption pilot plant can be constructed by applying temperature swing adsorption (TSA) technology. TSA involves low-temperature adsorption and high-temperature desorption, possibly combined with a pressure decrease (TVSA: temperature–vacuum swing adsorption [50]). The simplified process diagram of a fixed-bed adsorption plant with TSA technology is shown in Figure 12.

Another possibility is the choice of the correct reagent to be used in scrubbers for the abatement of acidic substances; the use of substances such as calcium carbonate and hydrated lime would lead [24,25], based on stoichiometric considerations, even to a partial reduction of the $CO_2$ generated.

In order to guarantee a win–win strategy, modification of the traditional structure of the flue gas depuration line must be verified and evaluated through an overall environmental assessment using tools such as Life Cycle Assessment (LCA) and traditional material and energy balances [51,52].

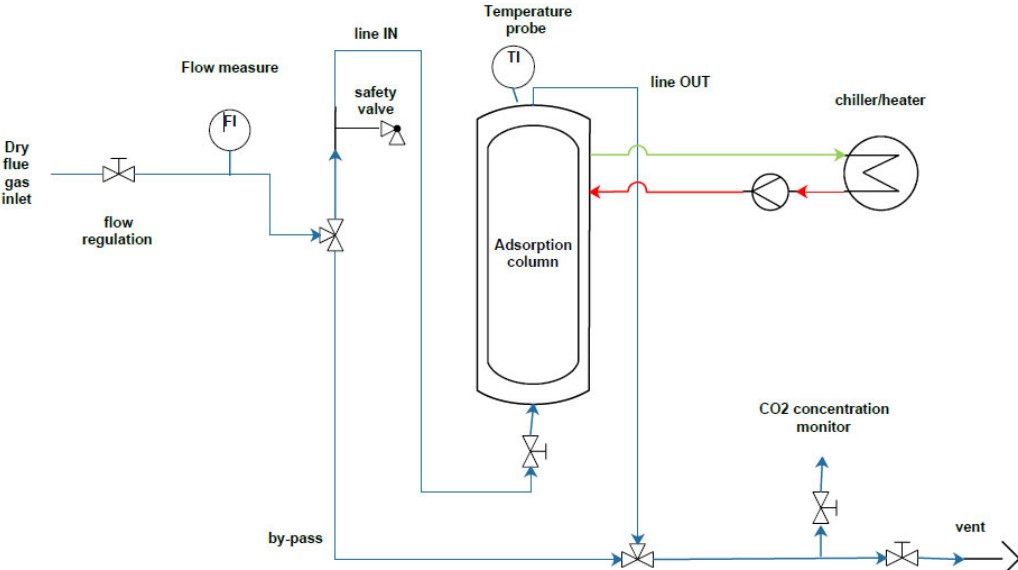

**Figure 12.** Simplified scheme of a temperature swing process for $CO_2$ adsorption.

## 4. Conclusions

To better understand the possible environmental impacts of these kinds of technologies, the flue gas emissions from thermal treatment were examined in this review paper. A thorough analysis of the thermal treatment given to the MSW matrix was carried out in the first phase. The second section analyzed the residual pollutant concentration (after the flue gas depuration system) and compared it to other industrial facilities as well as to the law as it stands now.

The $CO_2$ parameter received special attention. One of the greenhouse gases (GHG) contributing to the phenomenon of climate change is $CO_2$, a gas that is persistent in the atmosphere. Specific indicators ($tCO_2$/t MSW and $tCO_2$/MWh) were developed in this regard, taking into account the thermal treatment plants located in six Italian regions. A discussion followed about the potential decrease of these parameters, utilizing literature data.

The main obtained results show the following:

- From the primary pollutant parameters (dust, NOx, SOx, acid gases, and so on) in comparison with other industrial activities and with other waste management modalities, direct combustion (incineration) presents low emissions (for all the considered parameters).
- From the $CO_2$ parameter, the amounts coming from these kinds of plants are equal to more or less 0.86 $tCO_2$/tMSW and 0.44 $tCO_2$/MWh. These amount (expressed in terms of indexes) are lower than those coming from other kinds of management (for example for the landfill, the index is equal to 3.28 $tCO_2$/MSW).
- There are some technological possibilities to reduce the amount of $CO_2$ generated from these kinds of plants. However, these possibilities are at early stages of implementation, and so the main limitation is that there are no reduction technologies at the industrial scale. Future work is needed in order to improve this aspect.

**Author Contributions:** Conceptualization, D.P., M.R. and M.Z.; methodology, D.P., M.R. and M.Z.; software, D.P., M.R. and M.Z.; validation, D.P., M.R. and M.Z.; formal analysis, D.P., M.R. and M.Z.; investigation, D.P., M.R. and M.Z.; resources, D.P., M.R. and M.Z.; data curation, D.P., M.R. and M.Z.; writing—original draft preparation, D.P., M.R. and M.Z.; writing—review and editing, D.P., M.R. and M.Z.; visualization, D.P., M.R. and M.Z.; supervision, D.P., M.R. and M.Z.; project administration, D.P., M.R. and M.Z.; funding acquisition, D.P., M.R. and M.Z. All authors have read and agreed to the published version of the manuscript.

**Funding:** This research received no external funding.

**Institutional Review Board Statement:** Not applicable.

**Informed Consent Statement:** Not applicable.

**Data Availability Statement:** Not applicable.

**Conflicts of Interest:** The authors declare no conflict of interest.

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
