# Peer review of "An Overview of Thermal Treatment Emissions with a Particular Focus on CO2 Parameter"

_sustainability, doi:10.3390/su142315852_

Round 1
Reviewer 1 Report (Previous Reviewer 1)
Dear Authors,
Thank you for resubmission the manuscript.
Nevertheless, the following improvement need to be made:
1. CO2 is one of the main focuses of the paper. I suggest that more attention should be paid to this aspect. It is necessary to highlight new and unique points revealed through research, to find insight by analyzing previous studies.
2. I would suggest shortening the description of common processes (section 2) and to focus mostly on the main aspects of the paper (CO2 emissions and CO2 reduction)
3. The title should be changed to reflect more clearly the contribution of the study
4. Please explain some of the main review outcomes in the abstract and conclusion sections
5. Please mention some of the limitations associated with the review study conducted.
Author Response
REVIEWER 1
Dear Authors,
Thank you for resubmission the manuscript.
Nevertheless, the following improvement need to be made:
- CO2 is one of the main focuses of the paper. I suggest that more attention should be paid to this aspect. It is necessary to highlight new and unique points revealed through research, to find insight by analyzing previous studies.
Dear Reviewer, thanks a lot for your comments. In according to your indications we added some more explanation in the introduction sector.
- I would suggest shortening the description of common processes (section 2) and to focus mostly on the main aspects of the paper (CO2 emissions and CO2 reduction)
Dear Reviewer, thanks a lot for your comments. In according to your indications, we shortening the description of the thermal treatment process and add some more explanation about the point of CO2 concentration problem.
- The title should be changed to reflect more clearly the contribution of the study
Dear Reviewer, thanks a lot for your comments. We modified the paper title. The new title is “An overview of thermal treatment emissions with a particular focus on CO2 parameter”.
- Please explain some of the main review outcomes in the abstract and conclusion sections
Dear Reviewer, thanks a lot for your comments. We added the main review outcomes in the abstract and in the conclusion sections.
- Please mention some of the limitations associated with the review study conducted.
Dear Reviewer, thanks a lot for your comments.

Reviewer 2 Report (Previous Reviewer 3)
the authors did all the necessary corrections
Author Response
REVIEWER 2
The authors did all the necessary corrections
Dear Reviewer, thanks a lot for your comments.

Reviewer 3 Report (New Reviewer)
The study undertaken by the authors is well aligned with the scope of the journal and comprises of novelty in its approach. However, there are a few alterations which may improve the overall quality of work

Author Response
REVIEWER 3
The study undertaken by the authors is well aligned with the scope of the journal and comprises of novelty in its approach. However, there are a few alterations which may improve the overall quality of work:
- Figure 2 presented in the introduction consists of graphs plotted by employing data from 2017. Update the references that have been used for plotting the graph as the waste composition has changed drastically over the years.
Dear Reviewer, thanks a lot for your comment. We re-elaborated the Figure used the last data available.
- The entire manuscript consists of sentences which are very lengthy and complex. The sentences need to reframed and simplified in order to clearly convey the information to the readers.
Dear Reviewer, thanks a lot for your comment. We tried to simplify the sentences.
- Line 117: “A particular focus on CO2 parameter has been performed” – The objectives mentioned in the manuscript are not clear. It would be better if the objectives can be reframed and presented in a more lucid manner.
Dear Reviewer, thanks a lot for your comment. In according to your indications we modified and clarified it in the introduction.
- Line 160: “To facilitate chemical processes, air is typically employed, and it is supplied in excess of the stoichiometric amount” – Provide explanation for the provision of excess air in the incineration chamber as it will provide a much better understanding of the working of the incineration chamber.
Dear Reviewer, thanks for your comment. We provided some more explanation about this point (from line 186).
- Line 277: “As in the past, road transportation is confirmed as the main contributor to NOx, mainly from diesel engines” – Provide appropriate references to support the above-mentioned fact.
Dear Reviewer, thanks a lot for your comment. We added the reference [35]
- Line 298: “The generation plant runs in a co-generative mode at the moment.” – Explanation should be provided on how and why the plant are operating on a cogenerative mode.
Ok, thanks for the comment. When the Turin incineration plant was built started to work in only electric configuration (so with the production only of electric energy) and this because there was not the district heating network. The plant anyway was authorized to work in cogeneration configuration (so with the production both of electric and thermal energy). Currently the district heating network has been built and the plant works in cogeneration.
- Line 316: “As we know, environmental pollution is definitely a negative externality.” – Although various measures and provisions have been provided, they do not discuss in detail how the particular provision or measure aids in reducing the carbon emissions.
Thanks Mr Reviewer. The main provision is increase of the cost associate to the release of CO2 (€ / tCO2 released).
- Line 347: “In particular, an increase in temperatures and irradiation conditions could increase the concentrations of ground-level ozone and secondary pollutants.” – 2 Provide more detailed explanation on how the increase in temperature and irradiation conditions increases the concentration of the ground-level ozone and secondary pollutants.
Thanks dear Reviewer for your comment. We added the reference for this sentence [26]
- Line 407: “These values… 3.28 t CO2/t MSW.” – Provide reasoning for the lower value obtained as compared to other MSW management methods.
Thanks dear Reviewer for your comment. We did it in the paper text.
- Line 477: “The main obtained… (for all the considered parameters).” – The conclusion of the review can talk more about how the core objectives of the review have been met and provide relevant discussions on the future aspect of the current work.
Thanks dear Reviewer for your comment. We did it in the conclusion.

Reviewer 4 Report (New Reviewer)
This is a very informative review paper on the environmental effects of thermal treatment methods.
I have two main comments. First, I may suggest the authors be more specific in the title and include that the focus is on CO2 emission.
Also, please be more specific in describing the objectives (lines 123-125). What are the aimed gaps?
Author Response
REVIEWER 4
This is a very informative review paper on the environmental effects of thermal treatment methods.
I have two main comments.
First, I may suggest the authors be more specific in the title and include that the focus is on CO2 emission.
Dear Reviewer, thanks a lot for your comments. We modified the paper title. The new title is “An overview of thermal treatment emissions with a particular focus on CO2 parameter”.
Also, please be more specific in describing the objectives (lines 123-125). What are the aimed gaps?
Dear Reviewer, thanks a lot for your comments. We added this in the introduction sector. In particular this is the new modified sentence: “In this review the flue gas emissions from thermal treatment are analyzed in order to better understand the potential environmental impact of these kinds of processes. In the first part a complete review of the thermal treatment applied to MSW matrix has been performed. In the second part the residual pollutant concentration (after the flue gas depuration system) have been analyzed and compared to other industrial plants and with the current legislation. Furthermore, a particular focus on CO2 parameter has been performed. The CO2 is a persistent atmospheric gas and it is one of the Greenhouse Gas (GHG) potentially responsible of the Climate Change phenomena. As previously indicated for the CO2 there are no consolidated reduction technologies and no limit concentrations to be respect. Moreover generally the concentration of this parameters are not measured (and this because there are not limit concentrations to respect). In literature there are some papers that estimate these amount [27, 28]. In order to limit this gaps in this review a specific index (tCO2/tMSW and tCO2/MWh) has been elaborated considering the thermal treatment plants present in 6 Italian regions and afterward some considerations, using literature, concerning the possible reduction of these parameters has been reported”.

Round 2
Reviewer 1 Report (Previous Reviewer 1)
The authors did all the necessary corrections
This manuscript is a resubmission of an earlier submission. The following is a list of the peer review reports and author responses from that submission.
Round 1
Reviewer 1 Report
Thank you for submitting your manuscript, but I am sorry to say that in present form the manuscript is not ready for the publication.
The most part of the article is mostly the basic information about the studied question. I suggest to add a detailed information about the previously performed studies, in order to clarify what is the difference between this study and previous studies
I suggest to add more details to the original authors research part (focus to CO2 emission)
Is is not clear if there any original authors research findings in the field of CO2 reduction possibilities. It is needed to provide more detailed information (probably with results of some laboratory of field tests)
The manuscript is not well-organized. The title and the manuscript structure are more suitable for a review article and should be rewrite according to the requirements to research article
The highlights do not precisely present the key contribution of this study. I recommend the authors to revise it accordingly
Reviewer 2 Report
I think this is more of an educational blog for extension and outreach explaining what the thermal treatment process is. The environmental aspect is too shallow. Lack of reviews of what research has been done in the field.
Reviewer 3 Report
The below issues must be addressed
1-provide more explanation on "social sustainability"
2-how we can improve the "social sustainability" state the strategies and appropriate solutions
3-"A number of inert materials (the bed) are kept in suspension (fluid) inside the combustion chamber of the fluidized bed system by an upward air current (which also acts as a comburent)." so, what must be the solution
4-Improve the quality of Figure 7
5-"Figure 8. Incidence of annual emissions of the main sectors of activity in Italy in 2000 and 2018 for the contaminants of greatest interest (processing from ISPRA data, 2020)." this must be a Table not Figure
6-"Recently, with the growing urgency to adopt long-term sustainable technological solutions, research is being directed towards processes involving the use of materials of natural origin..." provide an additional explanation for those technological solutions
7-add a section for current gaps and a further recommendation section